# Wheat Grains as a Sustainable Source of Protein for Health

**DOI:** 10.3390/nu15204398

**Published:** 2023-10-17

**Authors:** Dalia Z. Alomari, Matías Schierenbeck, Ahmad M. Alqudah, Mashael Daghash Alqahtani, Steffen Wagner, Hardy Rolletschek, Ljudmilla Borisjuk, Marion S. Röder

**Affiliations:** 1Department of Clinical Nutrition and Dietetics, Faculty of Applied Medical Sciences, The Hashemite University, P.O. Box 330127, Zarqa 13133, Jordan; 2Leibniz Institute of Plant Genetics and Crop Plant Research (IPK), Corrensstraße 3, OT Gatersleben, D-06466 Seeland, Germany; wagner@ipk-gatersleben.de (S.W.); rollet@ipk-gatersleben.de (H.R.); borisjuk@ipk-gatersleben.de (L.B.); roder@ipk-gatersleben.de (M.S.R.); 3CONICET CCT La Plata, La Plata 1900, Buenos Aires, Argentina; 4Biological Science Program, Department of Biological and Environmental Sciences, College of Art and Science, Qatar University, Doha P.O. Box 2713, Qatar; aalqudah@qu.edu.qa; 5Department of Biology, College of Science, Princess Nourah bint Abdulrahman University, P.O. Box 84428, Riyadh 11671, Saudi Arabia; mdalqahtani@pnu.edu.sa

**Keywords:** bread wheat, grain quality, baking quality, flour, GWAS, genetic biofortification, candidate genes

## Abstract

Protein deficiency is recognized among the major global health issues with an underestimation of its importance. Genetic biofortification is a cost-effective and sustainable strategy to overcome global protein malnutrition. This study was designed to focus on protein-dense grains of wheat (*Triticum aestivum* L.) and identify the genes governing grain protein content (GPC) that improve end-use quality and in turn human health. Genome-wide association was applied using the 90k iSELECT Infinium and 35k Affymetrix arrays with GPC quantified by using a proteomic-based technique in 369 wheat genotypes over three field-year trials. The results showed significant natural variation among bread wheat genotypes that led to detecting 54 significant quantitative trait nucleotides (QTNs) surpassing the false discovery rate (FDR) threshold. These QTNs showed contrasting effects on GPC ranging from −0.50 to +0.54% that can be used for protein content improvement. Further bioinformatics analyses reported that these QTNs are genomically linked with 35 candidate genes showing high expression during grain development. The putative candidate genes have functions in the binding, remobilization, or transport of protein. For instance, the promising QTN AX-94727470 on chromosome 6B increases GPC by +0.47% and is physically located inside the gene *TraesCS6B02G384500* annotated as Trehalose 6-phosphate phosphatase (T6P), which can be employed to improve grain protein quality. Our findings are valuable for the enhancement of protein content and end-use quality in one of the major daily food resources that ultimately improve human nutrition.

## 1. Introduction

Cereal grains are a major source of energy, carbohydrates, and dietary proteins. Notably, 41% of grains are used for human consumption, and up to 35% are used for animal feed. Increasing the use of plant-based foods to replace animal-based foods is one feasible strategy for reaching the goal of replacing plant protein with animal protein [1].

Bread wheat (*Triticum aestivum* L.) is a staple crop for an estimated 35% of the world’s human population providing a fifth of global food calories. By 2022, wheat was cultivated on an estimated 221 million hectares, making it the most widely grown crop worldwide. In the last year, the global production reach 781 million tones, and together with rice and maize, wheat is considered one of the big global staple cereals [2]. Wheat grain protein content (GPC) is considered the main source of vegetable protein in human diets, particularly in regions where animal protein is scarce or expensive [3,4,5]. Wheat has two textural classes: soft and hard. The two classes of wheat are used for different purposes as hard wheat is mainly used for making bread and pasta, while soft wheat is utilized for making biscuits and cakes [6].

One of the major causes of infection in humans is a lack of secondary immunity caused by protein-energy malnutrition (PEM). Marasmus (chronic wasting) or kwashiorkor (edema and anemia) are the two symptoms of acute PEM in infants [7]. In children with chronic PEM, cognitive growth is hampered [8]. In developing countries, protein-energy malnutrition is one of the leading causes of death, with pregnant women and young children being the most vulnerable [9].

Overall, wheat grain’s chemical composition makes it a staple crop with versatile uses in various food products worldwide. Wheat grain consists primarily of carbohydrates (70–75%), mainly in the form of starch, followed by protein (10–15%), fats (1–2%), vitamins (such as B-vitamins), minerals (like iron and magnesium), fiber (2–3%), and water (10–15%) [10,11]. On the food processing side, the GPC and starch content of wheat grains constitute a crucial factor determining the baking quality of flour [12,13,14]. Wheat proteins, particularly gluten, play a significant role in determining the dough’s strength, elasticity, and extensibility, which ultimately determine the quality of baked products [15]. The gluten proteins contribute to the structure of bread by forming a strong and elastic network that traps gases produced by yeast during fermentation, leading to a well-risen and airy bread. Wheat varieties with high GPC are generally preferred for bread-making because they produce doughs that are easier to handle and result in bread with better volume, texture, and crumb structure [16]. Since the positive association between GPC and baking quality is critical for producing high-quality flour and baked products, more genetic studies are necessary to deepen knowledge of how to improve protein contents [10,17], taking into consideration the fact that selecting varieties with higher GPC was indicated as one of the priorities among breeders [18,19].

Exploration of genetic resources is a valuable approach for identifying novel sources of variation for traits, quantitative trait loci (QTLs), and candidate genes for improving grain quality and grain yield-related characters [20,21]. At present, genetic regions controlling GPC in bread wheat have been recognized by performing a QTLs approach using biparental as well as diverse collections [22,23,24,25]. Some reviews have listed significant marker-trait associations (MTAs) in several chromosomes [26]. As an example, a GWAS analysis performed by [24] reported two QTLs related to GPC on 2B and 6A. Recent reports by [17] documented some multi-environment genetic markers associated with GPC on 3A and 3B, while [27], studying a diverse panel of 184 accessions, reported 23 significant QTLs distributed over the wheat genome. Moreover, [28] analyzed 394 multiparent advanced generation intercross population lines (MAGIC) and documented 12 QTLs explaining only a small amount of phenotypic variance (≤10%) on many chromosomes. Nonetheless, and due to wheat’s importance as a source of vegetable protein worldwide and the association of GPC with baking and milling quality, novel attempts are necessary to explore the genetic basis of GPC in bread wheat helping human health.

In the current study, a GWAS was implemented to analyze 369 European wheat genotypes for GPC over three years. Our objective was to detect candidate genes linked to GPC and, subsequently, elucidate how the genetic mechanism controls GPC variations and how it can be implemented to improve GPC and end-use quality. Here, we present many significant markers which are physically co-located inside candidate genes. The identified genes are mostly involved in protein synthesis and accumulation mechanisms in wheat grains that improve grain nutrient and end-use quality which are crucial for human health.

## 2. Materials and Methods

### 2.1. Plant Material

The hexaploid wheat germplasm used comprised 369 elite European registered varieties including 355 winter genotypes and 14 spring accessions described in [29,30]. The genotypes were mostly from Germany and France in addition to other European countries. Field experiments were conducted at the Leibniz Institute of Plant Genetics and Crop Plant Research (IPK, Gatersleben, Germany) during the years 2014–2017 (2014/2015, 2015/2016, and 2016/2017) where the whole set of the germplasm was sown at each year. A plot with a size of 2 × 2 m was used for each genotype with six rows spaced 0.20 m apart. More details were described in a previous study [31]. Standard agronomic wheat management practices were applied to the soil and plots.

### 2.2. Determination of Total Seed Protein

Seed protein was determined from dry mature seeds that were finely ground before analysis (Retsch mill MM200). Analysis was done using near infrared (NIR) spectroscopy (MPA, Multi Purpose Analyzer, Bruker GmbH, Bremen, Germany) [32], applying multivariate calibration algorithms (software OPUS, Bruker GmbH) and the reference material library B-FING-S (Bruker GmbH).

### 2.3. Statistical Analysis

The variance (ANOVA) analysis for protein was calculated, and the significant differences among genotypes and years were determined at a probability level of *p* ≤ 0.05. The Pearson correlation coefficient was used to evaluate the relationships among the measured parameters of data (*p*-value ≤ 0.05). ANOVA and Pearson’s correlation coefficient were calculated using GenStat v19 software [33]. Using GenStat v19, the best linear unbiased estimates (BLUEs) over three years were calculated from restricted maximum likelihood (REML) analysis with a mixed linear model and considering genotype as a fixed effect and the environment as a random effect. BLUEs were calculated for each genotype of each trait across the years (2015, 2016, and 2017).

Broad-sense heritability (*H*^2^) was calculated for each trait using the formula:*V*_G_/(*V*_G_ + ((*G*_e_/ nE))
where *V_G_* is the variance of the genotype, *G*_e_ represents the variance of the residual, and nE is the number of years [34].

### 2.4. Genotyping and Marker Quality Control

Our wheat population was genotyped using two marker arrays: a 90K iSELECT Infinium array including 7761 markers and a 35K Affymetrix SNP array (Axiom^®^ Wheat Breeder’s Genotyping Array) including 7762 markers [35,36] (https://www.cerealsdb.uk.net/cerealgenomics/). These two arrays were genotyped by using the SGS-TraitGenetics GmbH, Gatersleben, Germany (www.traitgenetics.com) described for this germplasm by [29,30]. The ITMI-DH population was used as a reference map [37,38] to anchor the SNP markers of the 90 K and 35 K arrays. To obtain high-quality makers, the SNPs in chips underwent a quality control and filtration process by removing the markers with ≥3% missing values, a minor allele frequency (MAF) of ≤3%, and markers with unknown chromosomal positions. Then, we used the physical position of the wheat genome sequence RefSeq v1.1 for the SNPs.

### 2.5. GWAS Analysis

GWAS analysis was calculated by using the Genomic Association and Prediction Integrated Tool (GAPIT 3) in R software [39,40]. First, GWAS analysis was computed by using the mixed linear model (MLM), which took into account the variance–covariance kinship matrix and PCA and was accomplished by incorporating the phenotypic and genotypic datasets. Moreover, the kinship matrix was calculated using the VanRaden method [41] to determine the relative kinship among the sampled individuals. Both PCA and the kinship matrix were used for population correction and stratification.

Another recent powerful GAPIT model known as the Fixed and random model Circulating Probability Unification (FarmCPU) was applied to our data analysis. The FarmCPU model was applied by considering the incorporation of multiple markers simultaneously as a covariate in a fixed effect model and optimization on the associated covariate markers in a random effect model separately, which empowered us to avoid any false negative and control the false positive associations by preventing model overfitting [42]. Thus, FarmCPU is a powerful tool with less false positives than MLM. The selection of an appropriate model and threshold are important steps in identifying markers that are truly associated with specific traits and which could be located within or very close to genes that control the trait variation, while controlling both false-positive and false-negative associations. To determine which of the tested models best fit the data, we plotted the quantile-quantile (Q-Q) plot which was drawn based on the observed and expected −log_10_ (*p*) values. Then, based on the GWAS output of the three models (GLM, MLM, and FarmCPU), the number of significant associations, and the resulted QQ-plot, we selected the FarmCPU. A threshold *p*-value ≤0.0001 equal to −log_10_ (*p*) ≥ 4 was considered to indicate significant associated quantitative trait nucleotides (QTNs) and used for further analysis. Marker effects (positive/negative on %GPC) and phenotypic variance explained by the associated markers (*R*^2^) were removed from the GWAS results.

### 2.6. Genes’ Identification, Annotation, and Expression Analysis

Significant markers and the markers located within the LD region (*r*^2^ ≥ 0.2) were considered for BLAST. The sequence of the identified makers was obtained from the wheat 90k [35] and 35k database [36]. Marker sequences were BLASTed against the recently released IWGSC RefSeq v1.1 genome by Ensemble Plants (http://plants.ensembl.org/Triticum_aestivum) to obtain their gene annotation. The expression profile of all the putative candidate genes associated with the identified SNPs was checked using the published RNA-seq expression database of wheat in the WheatGmap web tool (https://www.wheatgmap.org) [43].

## 3. Results

### 3.1. Variations of Grain Protein Content in a Worldwide Winter Wheat Panel

Grain protein content was significantly influenced by the years, genotypes, and *Year × Genotypes* interactions (*p <* 0.001). Data analysis revealed extensive phenotypic variation in all studied traits. Broad-sense heritability for GPC equaled 0.8. GPC among different years, and genotypes showing higher and lower GPC across the environments, summary statics, and correlations are exhibited in Figure 1 and Figure 2 and Appendix A.

Grain protein values ranged between 9.9–14.15% (2015), 9.7–13.12% (2016), 9.5–15.31% (2017), and 10.0 to 13.99% for BLUEs (Figure 1a; Appendix A). The genotypes showing the higher and lower GPC across the years are plotted in Figure 1b,c. Accessions such as Hamac (13.98%), Hereward (13.96%), Cassiopeia (13.77%), Incisic (13.65%), Levis (13.35%), Renan (13.13%), and Alidos (13.05%) stood out for their high GPC values (Figure 1b). For their part, accessions showing the lower GPC values were Graindor (9.92), Alchemy (9.95), Haussmann (9.96), Equilibre (9.99), Rosario (10.01), and Ambrossia (10.08) (Figure 1c). Low to high positive Pearson’s correlation coefficients were detected among environments and BLUEs, showing variation in GPC among genotypes and environments (Figure 2).

### 3.2. QTNs Underlying GPC Variations

Using the Farm-CPU method, 54 significant QTNs distributed over 12 chromosomes related to grain protein content were detected (−log_10_ > FDR; *p < 0.0001*) in three environments and BLUEs. These QTNs were reported in chromosomes 1A (5), 2B (1), 3A (10), 3B (6), 4A (1), 5A (4), 5B (1), 6A (12), 6B (8), 6D (1), 7A (4), and 7B (1) (Figure 3). Overall, 4 significant QTNs were reported in 2015, 26 in 2016, and 17 in 2017, along with 7 for BLUEs. Grain protein content effects ranged from −0.50 to +0.54% (Table 1). More details about the QTNs detected such as chromosome, marker position, effect on %GPC, −log_10_, target allele, related candidate gene, and candidate gene annotation are shown in Table 1.

### 3.3. High-Confidence Candidate Genes Related to GPC

Further analysis revealed 35 high-confidence candidate genes based on the QTNs’ positions (Table 1). These CGs are located in Chr. 1A (4), 2B (1), 3A (4), 3B (4), 4A 5A (1), 5B (1), 6A (9), 6B (5), 6D (1), 7A (4), and 7B (1).

In Figure 4, the biological functions, cellular components, and molecular functions associated with the reported candidate genes are indicated (GO enrichment analysis). In total, 32 of the analyzed CGs showed biological functions; 7 of them are involved in cellular components, and 20 have some molecular functions. Interestingly, the most important biological processes and molecular functions involved were related to disaccharide and oligosaccharide biosynthesis and metabolism.

Based on the GWAS results, the most reliable candidate genes were mined and selected due to QTNs showing positive effects on GPC, related gene networks, and high Ref-Seq expression level during grain development (Table 1, Figure 5, Appendix A Appendix A). Three QTNs (AX-94552678, BobWhite_rep_c64315_180 and AX-109292583) showing −log_10_ values ranging from 4.44 to 4.51 and positive effects on %GPC between 0.501 and 0.515% are inside *TraesCS5A02G429000*, located in chromosome 5A at 613543346-613547572 bp and annotated as Ubiquitin-conjugating enzyme/RWD-like (Table 1, Appendix A).

Several other important QTNs were reported on chromosome 6B. Four QTNs (RAC875_c18659_651, wsnp_Ku_c8343_14190318, wsnp_Ex_c8011_13584847 and wsnp_Ex_c13352_21044607) presented GPC effects ranging from 0.281% to 0.286% and −log_10_: 4.63–4.73. These markers were collocated within *TraesCS6B02G071500* at 48347795-48354269 bp, a gene encoding a TFIIS central domain-containing protein (Table 1, Appendix A).

The QTN AX_95155979 showing a −log_10_:4.73 and increases of 0.286% on GPC was reported inside *TraesCS6B02G071700* (48411470-48415030 bp), which encodes Sialyltransferase-like protein 2 (Appendix A). Another QTN (AX-94727470) (−log_10_: 4.34 and effect +0.472%) is located within *TraesCS6B02G384500* (659232852-659237118 bp)*,* annotated as Trehalose 6-phosphate phosphatase (Appendix A). For its part, AX-158588655 (−log_10_: 4.16 and effect +0.54%) was collocated with *TraesCS6B02G391200*, a gene encoding ATP-dependent RNA Helicase DDX51, located at 665512071-665516262 bp (Table 1, Appendix A).

On Chromosome 7B, the QTN AX-94830265 showing LOD: 4.23 and a positive effect on %GPC (0.525%) was located inside *TraesCS7B02G047600* at 47000524-47004008 bp, a gene annotated as Plasma membrane ATPase (Table 1, Appendix A).

The expression analysis of some of the CGs reported showed a wide range of gene expression. High expression was reported during grain development and interestingly also in flag leaf development during the grain filling period (Figure 5 and Appendix A). The highest transcription values in grain tissues were detected for *TraesCS5A02G429000*, *TraesCS5B02G303800*, *TraesCS6B02G071700*, *TraesCS1A02G314400,* and *TraesCS6B02G073600*. For flag leaf tissues, high expression was reported for *TraesCS6B02G073600*, *TraesCS5A02G429000,* and *TraesCS3B02G317300*. The rest of the CGs reported showed lower expression values ranging from 0.1 to 5 TPM (Appendix A).

## 4. Discussion

### 4.1. The Importance of GPC on Bread Wheat, the Main Source of Vegetable Protein Worldwide

Due to the importance of grain protein content for flour quality and the sub-products derived, this study focused on studying the variability of GPC in a European elite panel comprising 369 genotypes in three environments and the further detection of QTNs and candidate genes using 15523 valid markers through the Farm-CPU model. 

Our results indicate a wide variability in GPC in European genotypes (9.52–15.31%). Protein content in grains is largely influenced by genetics, water and nitrogen availability, biotic and abiotic stresses, and grain-filling duration [10,28,44]. These multiple factors explain the intermediate correlation with GPC reported for the years evaluated (0.17–0.72). Moreover, the high heritability reported for GPC was documented by [24] (0.88–0.91), [27] (0.56–0.82), [17] (0.68–0.79) and indicates the usefulness of this set for GWAS studies.

In a recent study, [17], evaluating a 255 worldwide winter panel, documented that GPC fluctuated from 8.6 to 16.4%. For their part, [27], evaluating 184 diverse genotypes under Indian conditions, reported GPC variations ranging from 8.6 to 15.81%. In this sense, our work has documented the variation in GPC in one of the largest European winter wheat sets analyzed so far. Geyer et al. [28] reported a GPC variability of 11.8–16.2% on a MAGIC population of 394 lines. Our results indicate that some accessions such as Hamac, Hereward, Cassiopeia, Incisic, Levis, Renan, and Alidos showed high and stable GPC among the environments (13–14%), while others showed low GPC values (9.9–10%) (Graindor, Alchemy, Haussmann, Equilibre, Rosario, and Ambrossia). In this sense, [28,45] suggest that genotypes with moderate to high GPC (>12%) are required for the production of bread, while those with low GPC are desirable for other purposes such as cookies, noodles, and animal feed. However, there are important variations in quality standards between countries.

### 4.2. Novel Candidates’ Genes with High Effect on GPC and High Expression during Grain Filling

In the analysis performed in the current work, 54 significant QTNs related to grain protein content were detected using −log_10_ > FDR (*p* < 0.0001) in three environments and BLUEs. These QTNs were reported in chromosomes 1A (5), 2B (1), 3A (10), 3B (6), 4A (1), 5A (4), 5B (1), 6A (12), 6B (8), 6D (1), 7A (4), and 7B (1). Further analysis revealed 35 high-confidence candidates’ genes collocated with most of the QTNs reported.

As was previously reported, GPC is quantitative trait influenced by numerous environmental factors and controlled by several genes [17,22,23,24,25,26,27,28]. Previous reports related to QTLs involved in GPC in all wheat chromosomes used a GWAS approach, although reports for CPG in European genotypes have not been studied in depth, highlighting the importance of our study as indicated previously. Even though other studies have documented markers associated with GPC in chromosomes 1A, 2B, 3A, 3B, 4A, 5A, 5B, 6A, 6B, 6D, 7A, and 7B, only some of the QTNs here documented have been previously reported in the literature. Coinciding with our results, [24] report some QTLs affecting grain starch content (QGsc.ipk-3A, QGsc.ipk-6A) and GPC (QGpc.ipk-6A) that are located with CAP11_c6193_232, AX-94973054 and Tdurum_contig46828_730 QTNs, respectively. These novel results would indicate their potential use for the improvement of these physiological characteristics.

Our results exhibited some QTNs (AX-94552678, BobWhite_rep_c64315_180, and AX-109292583) located inside *TraesCS5A02G429000* on 613543528–613544399 bp, a gene annotated as Ubiquitin-conjugating enzyme/RWD-like. According to different authors, Ubiquitination regulates varied plant growth and developmental processes related to protein translocation within cells, cell cycle control, photomorphogenesis responses to abiotic/biotic stress, floral development, hormone signaling, proteome homeostasis regulation, and signaling [46,47]. Interestingly, Refs. [48,49] reported the effect of ubiquitin pathways controlling seed size and weight in wheat and other species (Appendix A). Recent reports documented that the here-reported *TraesCS5A02G429000* gene also plays a role in the chlorophyll content of flag leaves at different grain filling stages under different phosphorus supplies on two double haploid populations (Jinmai 47 × Jinmai 84) and (Jinmai 919 × Jinmai 84) [50]. These results are in line with the high expression levels in flag leaf and grain tissues during post-anthesis (>45 TPM) and the positive effects of this candidate gene on GPC (+0.5%), suggesting its potential utility in breeding programs (Table 1, Figure 5 and Appendix A).

We reported several QTNs and candidate genes in chromosome 6B. For instance, four QTNs (RAC875_c18659_651, wsnp_Ku_c8343_14190318, wsnp_Ex_c8011_13584847, and wsnp_Ex_c13352_21044607) presented medium effects on GPC (*ca.* +0.29%). These markers were collocated within *TraesCS6B02G071500* at 48347795-48354269 bp. A transcription elongation factor S-II (TFIIS) has been related to this candidate gene, and no previous reports have delved into the effect of TFIIS on wheat GPC. In *Arabidopsis*, a deficient TFIIS mutant showed normal growth but deficient seed dormancy, suggesting its role in seed development [51]. Other recent studies suggest that the *Arabidopsis* TFIIS mutant proved highly sensitive to heat stress [52]. Moreover, the absence of the ET1 protein (similar to TFIIS) in mutants affects the starch synthesis and consequently endosperm development in maize seeds [53]. In view of these results, further studies should investigate the role of TFIIS in protein accumulation in wheat grains.

The QTN AX-95155979 (48414777 bp) increased GPC up to 0.29% and was collocated with the gene *TraesCS6B02G071700*, which encodes for Sialyltransferase-like protein 2 (SIA2) in the position 48411470-48415030 bp. Although the effect of this gene has not been previously documented in wheat, the functions associated with SIA2 have been reported. In *Arabidopsis*, the effect of the SIA2 protein has been associated with the stability of the pollen tube cell wall [54]. Moreover, mutations on SIA2 such as MALE GAMETOPHYTE DEFECTIVE 2 lead to abnormal pollen tube phenotypes [55].

Another QTN (AX-94727470) (effect +0.472%) is located within *TraesCS6B02G384500* (659234876 bp), annotated as Trehalose 6-phosphate phosphatase (T6P) (Table 1, Appendix A). As an intermediate of the trehalose metabolic pathway, T6P plays a key role for plant sugar signaling as well as the regulation of plant growth and development [56,57]. Moreover, [58] indicated that this signaling system is a key mechanism of resource allocation related to assimilating partitioning and grain yield improvement in several crops. The relationship with the T6P/SnRK1 regulatory system plays a role in controlling whole-plant resource allocation and source-sink interactions in crops. A recent report described several functions of Trehalose-6-phosphate phosphatase genes (TPPs) in wheat development and stresses response using a GWAS approach [59]. Interestingly, exogenous applications of different T6P precursors from 5 to 20 after anthesis on wheat under control and drought conditions showed a delay in plant senescence, increased crop resilience, higher chlorophyll content, and increases in grain size as well as grain starch and protein concentration [57]. Under *Mizus persicae* (green peach aphid) infections in *Arabidopsis*, Trehalose phosphate synthase 11 (TPS11) promotes the re-allocation of carbon into starch at the expense of sucrose, the primary plant-derived carbon and energy source of this pest, reducing the severity produced by this aphid infestation [60]. These previous reports highlighting the functionality of this gene in source/destination relationships would suggest the importance of this CG as a key factor in increasing the genetic gain on GPC.

On Chromosome 7B, the QTN AX-94830265 (47000619 bp) showing effects on GPC up to 0.525% is located inside *TraesCS7B02G047600* at 47000524-47004008 bp. This gene is annotated as Cation transporting Plasma membrane ATPase (P-Type ATPase). Although no previous reports of this QTN and CG were found in wheat, various authors have documented that P-Type ATPase superfamily genes play various roles in plant growth and development, biotic and abiotic responses, and hormonal signaling in rice, *Arabidopsis* [61,62], and soybean [63] (Appendix A). In wheat, 42 P-type II Ca^2+^ATPase genes were reported by [64]. A further analysis developed by these authors suggests their role in diverse functions related to growth and development, cell division, pollen tube growth, and biotic/abiotic tolerance. Coinciding with our results, high expression of these genes was reported during grain filling in grains and leaves in wheat (Appendix A), suggesting its utility as a candidate gene explaining GPC in this crop.

The QTN AX_158588655 (showing increases of +0.55% on GPC) is collocated with *TraesCS6B02G391200*, a gene encoding ATP-dependent RNA Helicase DDX51 at 665512071-665516262 bp (Appendix A, Table 1; Appendix A). The role of *TraesCS6B02G391200* as a source of resistance to *Septoria tritici* blotch at seedling and adult stages on 377 advanced spring wheat breeding lines from ICARDA has been reported [65]. A review by Liu and Imai [66] highlights the pleotropic role of RHs ATPases in *Arabidopsis*, maize, and rice due to their role in pre-rDNA processing. Knockdown mutants of different RHs (*OsBIRH1* and *OsSUV3*) showed differential effects on biotic and abiotic tolerance in rice [67,68]. In *Arabidopsis*, the genes *HEN2* and *AtMTR4* exhibited effects on plant and flower development [69,70]. Other important functions were reported for *AtRH57* (Glucose and ABA response), *AtRH3* (Chloroplast development), and AtRH22 (seed oil biosynthesis) [66]. In wheat, an ATP-dependent DNA helicase (*TaDHL-7B*) was reported and related to a novel reduced-height (*Rht*) gene reducing plant height without a grain yield penalty using QTL and GWAS analysis [71]. These previous reports suggest the need to deepen the studies of this candidate gene as a source to modify the GPC in wheat.

## 5. Conclusions

Grain protein content has been indicated as a crucial trait for grain end-use quality for food scientists and nutritionists. In our study, a wide range of GPC in bread wheat was measured, and candidate genes which subsequently elucidate the genetic mechanism underlying the high nutritional value of grains were reported using a 90k iSELECT Infinium and 35k Affymetrix arrays with the FarmCPU method. These CGs showed a wide range of expression in grain during the filling period. Moreover, from the human nutritional perspective, our results can improve the recommended protein intake. Our results provide new insight into the genetic control of GPC and would be of interest for enhancing wheat end-use quality and nutritional value.

## Figures and Tables

**Figure 1 nutrients-15-04398-f001:**
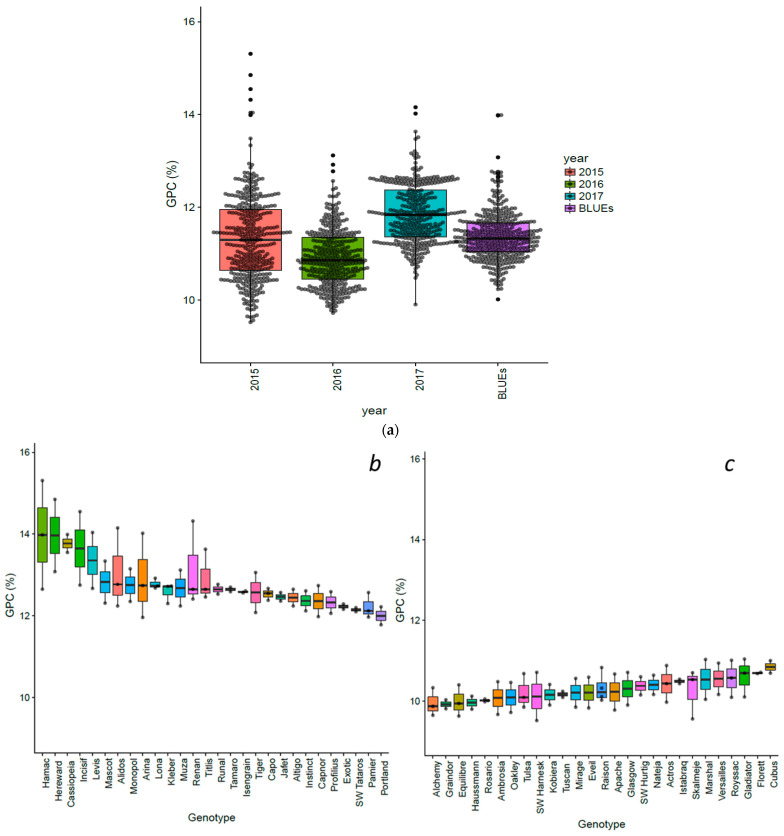
(**a**) Boxplot and jitter for three years and BLUEs values for grain protein content (GPC) in 369 European elite wheat genotypes. Genotypes showing (**b**) higher and (**c**) lower %GPC values across the years evaluated.

**Figure 2 nutrients-15-04398-f002:**
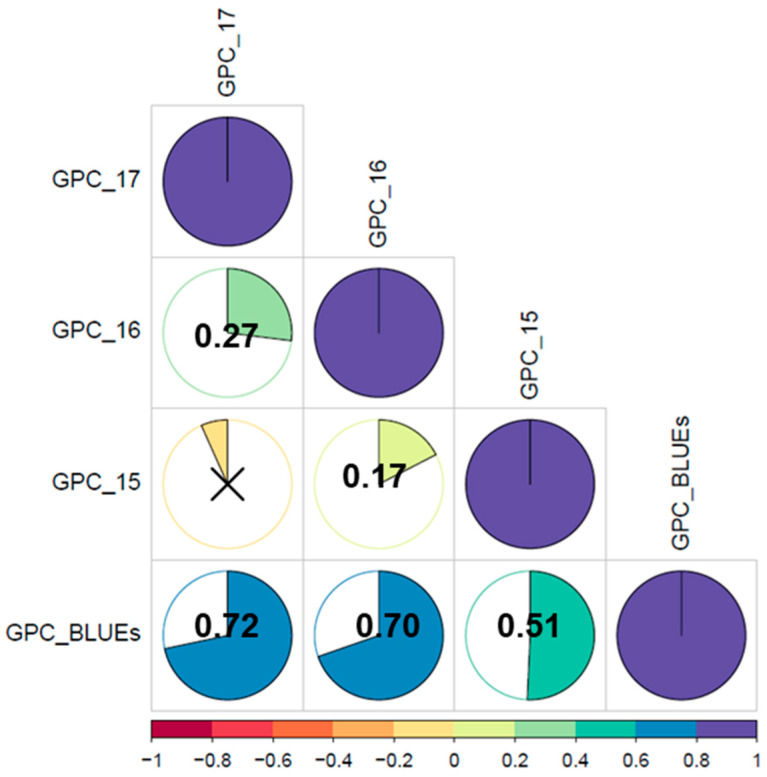
Correlation among grain protein in 369 winter wheat genotypes evaluated in four environments. The degree of significance for all correlations across different years was *p < 0.001*. The color reflects the strength of the correlation. Non-significant correlations are expressed using crosses.

**Figure 3 nutrients-15-04398-f003:**
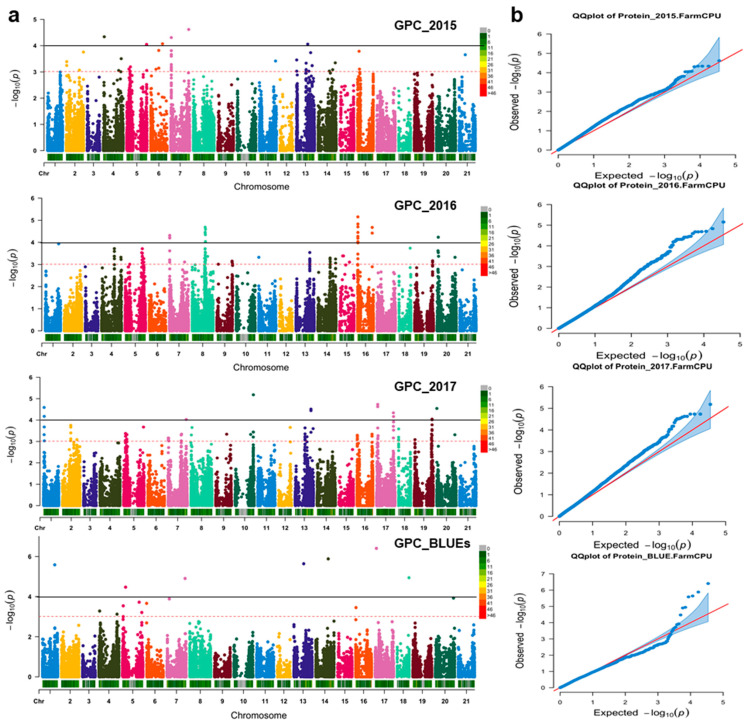
(**a**) Manhattan plots showing significant marker traits association for agronomical traits in 369 winter wheat genotypes in three environments and BLUEs values (*p* < 0.0001; −log_10_ > FDR). (**b**) qq plots for grain protein concentration; red line represents the expected values.

**Figure 4 nutrients-15-04398-f004:**
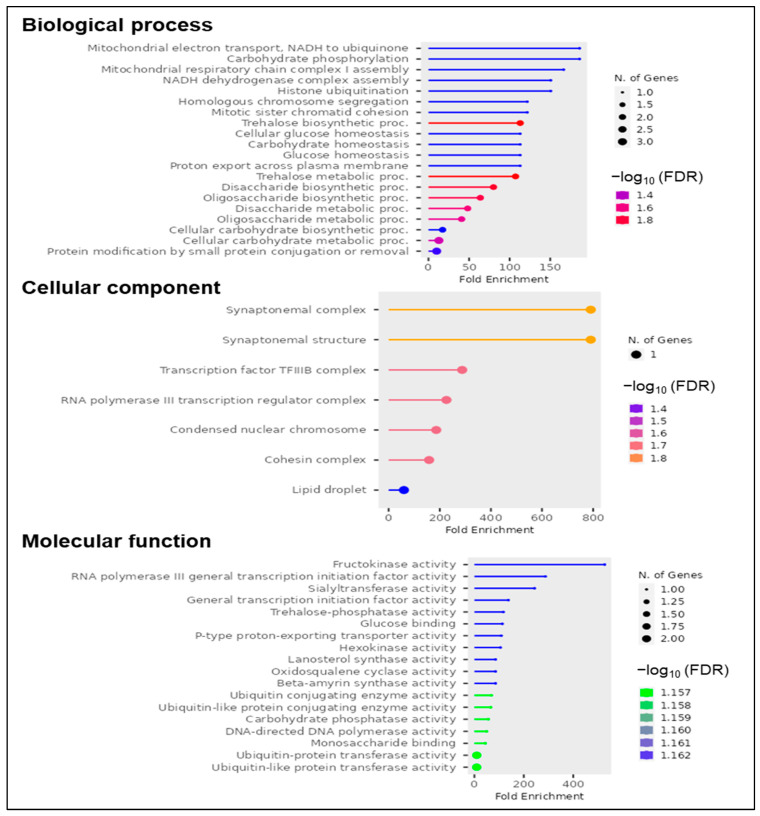
Gene ontology (GO) enrichment analysis.

**Figure 5 nutrients-15-04398-f005:**
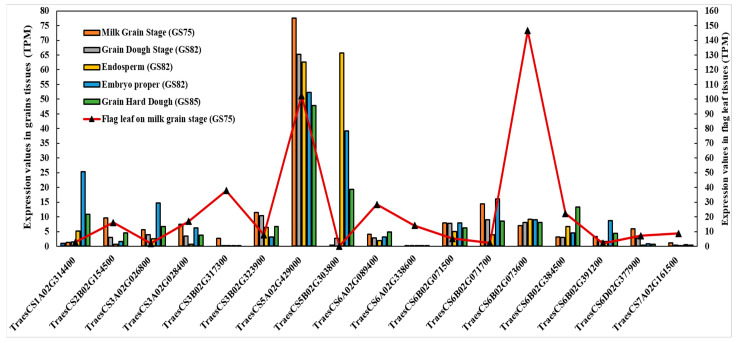
Expression value TPM (Transcripts Per Kilobase Million) of candidate genes in flag leaf (during milk grain stage) and grain development (Milk grain stage, dough stage, endosperm in dough stage, embryo proper in dough stage and grain hard dough).

**Table 1 nutrients-15-04398-t001:** Significant QTNs and candidate genes associated with grain protein content of 369 genotypes analyzed in three environments and BLUEs through the FarmCPU model.

Env	Chr	Marker	Effect (%GPC)	−log10	Position (bp)	Candidate Gene and Genomic Location (bp)	Annotation (Superfamily and PANTHER)
**2017**	1A	AX-94392216	−0.344	4.59	22611655	*TraesCS1A02G041100* (22612227-2614080) RGA5 gene	P-loop containing nucleoside triphosphate hydrolase
**2017**	1A	AX-158560740	−0.266	4.17	27275836	NA	
**blue**	1A	AX-158556547	0.119	5.58	476981928	*TraesCS1A02G279600* (476972557-476981741)	Josephin domain
**2015**	1A	RAC875_c46551_339	0.172	5.04	506283718	*TraesCS1A02G314400* (506281816-506286607)	Homeobox-like domain superfamily
**2015**	1A	IAAV6234	0.154	4.11	513893374	*TraesCS1A02G323500* (513879955-513894399)	P-loop containing nucleoside triphosphate hydrolase; DNA/RNA polymerase superfamily
**blue**	2B	AX-158536988	−0.123	4.47	122713731	*TraesCS2B02G154500* (122710128-122714573)	Protein Rolling Stone-like
**2016**	3A	AX-94451685	0.240	4.22	14045695	*TraesCS3A02G026800* (14045083-14049309)	DEK C-terminal domain
**2016**	3A	AX-94486651	0.240	4.31	14045732		
**2016**	3A	Excalibur_c10383_432	0.240	4.31	14047699		
**2016**	3A	Excalibur_c11505_155	0.240	4.31	14850594	*TraesCS3A02G027700* (14848772-14852646)	Tetratricopeptide-like helical domain superfamily
**2016**	3A	RAC875_c20134_535	0.240	4.31	14851011		
**2016**	3A	IAAV1155	0.240	4.31	14851251		
**2016**	3A	Excalibur_c92401_157	0.240	4.31	15089050	*TraesCS3A02G028300* (15086445-15089436)	Alpha-ketoglutarate-dependent dioxygenase AlkB-like superfamily
**2016**	3A	CAP11_c6193_232	0.239	4.20	15090085	*TraesCS3A02G028400* (15089868-15092852)	A0A077RAM9 (hypothetical protein wheat)
**blue**	3A	BS00065734_51	0.228	4.90	711095135	NA	
**2017**	3A	BS00065734_51	0.485	4.03	711095135	NA	
**2016**	3B	AX-108848182	−0.213	4.46	511035835	*TraesCS3B02G317300* (511034602-511051546)	Peptidase S8/S53 domain superfamily
**2016**	3B	AX-158537019	−0.220	4.69	511074018	*TraesCS3B02G317600* (511072080-511076195)	Galactose-binding-like domain superfamily
**2016**	3B	AX-111060338	−0.214	4.42	511507665	NA	
**2016**	3B	AX-158538466	−0.214	4.35	519416654	*TraesCS3B02G320500* (519415064-519417176)	ATPase, nucleotide binding domain
**2016**	3B	AX-158558088	−0.204	4.04	522280255	NA	
**2016**	3B	AX-110467694	−0.220	4.59	524450613	*TraesCS3B02G323900* (524449173-52445430)	UDP-Glycosyltransferase/glycogen phosphorylase HAD-like superfamily
**2017**	4A	AX-108845109	0.642	5.18	712225082	NA	
**blue**	5A	AX-158542530	0.156	5.64	382113600	NA	
**2017**	5A	AX-94552678	0.501	4.51	613543528	*TraesCS5A02G429000* (613543346-613547572) UBC2 Gene	Ubiquitin-conjugating enzyme/RWD-like
**2017**	5A	BobWhite_rep_c64315_180	0.501	4.51	613543528		
**2017**	5A	AX-109292583	0.515	4.44	613544399		
**blue**	5B	AX-158525605	−0.156	5.88	488112608	*TraesCS5B02G303800* (488111479-488113567)	Polyketide synthase, enoylreductase domain
**2016**	6A	AX-158552362	0.305	4.19	10493939	*TraesCS6A02G021300* (10491634-10496460)	Sam-dependent Methyltransferase
**2016**	6A	AX-108894863	0.313	4.27	10494137		
**2016**	6A	RAC875_c22627_315	0.295	5.15	10560290	*TraesCS6A02G021600* (10559982-10561177)	Uncharacterized protein
**2016**	6A	AX-95007092	0.301	4.45	11965153	*TraesCS6A02G024000* (11963972-11966414)	C2H2 zinc finger transcription factor
**2016**	6A	AX-111512288	−0.277	4.70	12058265	*TraesCS6A02G024100* (12055217-12058516)	LRRNT_2 domain-containing protein
**2016**	6A	AX-110469066	−0.287	4.83	12078919	*TraesCS6A02G024200* (12077818-12079854)	OS10G0469600 PROTEIN
**2016**	6A	AX-158588344	−0.262	4.03	12312937	*TraesCS6A02G024800* (12313087-12314289)	F-box domain-containing protein
**2015**	6A	BS00073124_51	−0.220	4.19	57728595	*TraesCS6A02G089400* (57725544-57732746)	Calcium-dependent protein kinase 16
**2016**	6A	AX-158530854	−0.456	4.42	571851398	*TraesCS6A02G338300* (571851256-571855462)	E3 ubiquitin ligase
**2016**	6A	AX-110545207	−0.456	4.42	571851707		
**2016**	6A	AX-94973054	−0.502	4.68	571852988		
**2016**	6A	Tdurum_contig46828_730	−0.502	4.68	571929129	*TraesCS6A02G338600* (571928076-571931078)	Aminotran_1_2 domain-containing protein
**2017**	6B	RAC875_c18659_651	0.281	4.63	48348762	*TraesCS6B02G071500* (48347795-48354269)	Transcription elongation factor S-II, central domain superfamily
**2017**	6B	wsnp_Ku_c8343_14190318	0.281	4.63	48348762		
**2017**	6B	wsnp_Ex_c8011_13584847	0.286	4.73	48349076		
**2017**	6B	wsnp_Ex_c13352_21044607	0.286	4.73	48352481		
**2017**	6B	AX-95155979	0.286	4.73	48414777	*TraesCS6B02G071700* (48411470-48415030)	Sialyltransferase-like protein 2
**blue**	6B	CAP7_rep_c6771_332	0.146	6.41	49984647	*TraesCS6B02G073600* (49984316-49986442)	Protein lurp-one-related 1-related
**2017**	6B	AX-94727470	0.472	4.34	659234876	*TraesCS6B02G384500* (659232852-659237118) TPS11 gene	Trehalose 6-phosphate phosphatase
**2017**	6B	AX-158588655	0.539	4.16	665512073	*TraesCS6B02G391200* (665512071-665516262)	ATP-dependent RNA Helicase DDX51
**blue**	6D	RAC875_c64099_90	−0.105	4.94	460570647	*TraesCS6D02G377900* (460567567-460573565)	Protein lutein deficient 5, chloroplastic
**2017**	7A	AX-95203767	0.348	4.54	1281744	*TraesCS7A02G001700* (1280349-1295461)	Terpene cyclase/mutase family member
**2015**	7A	wsnp_Ra_c4418_8012732	0.293	4.51	118156309	*TraesCS7A02G161500* (118145757-118159311)	PPR_long domain-containing protein
**2017**	7A	BS00022169_51	0.283	4.04	691259601	*TraesCS7A02G501500* (691258415-691260048)	OS10G0469600 Protein
**2017**	7A	AX-158567041	0.276	4.01	691474553	*TraesCS7A02G502400* (691472525-691473807)	Peptidase A1 domain-containing protein
**2016**	7B	AX-94830265	0.525	4.23	47000619	*TraesCS7B02G047600* (47000524-47004008)	Plasma membrane ATPase

## Data Availability

The original contributions presented in the study are included in the article/Appendix A. The genotypic data of 90K and 35K gene-associated SNPs used in this study were published in [35] (90K chip) and [36] (35K chip). Further inquiries can be directed to the corresponding authors.

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
