# Peer review of "Wheat Grains as a Sustainable Source of Protein for Health"

_nutrients, 2023, doi:10.3390/nu15204398_

Round 1

Reviewer 1 Report

General comments:  Needs to be formatted correctly for review.  Line numbering ends on page 5. 

Line 27:  Change “was designed to discover” to “focused on”

Line 28:  Change “identify” to “identified”

Line 31:  Change “worthy” to “significant”

Line 46:  Reword as “…aestivum L.), a staple…”

Lines 49-50:  None of these citations are in the reference list.

Line 56:  Citation missing from reference list.

Line 58:  Change to read “…factor for determining the…”

Line 67:  Change to read “…is critical for producing…”

Line 75:  Change “Up to now” to “At present”

Line 83:  Change “analyzing” to “analyzed”

Line 84:  Add “and” before “documented”

Line 93:  Add “are” before “physically”

Line 100:  There is no Alomari et al. 2019 citation in the reference list.  Should there be another  publication listed there or do you mean 2017 or 2018?  Bad mistake since this is a self citation.  Says you did a very poor job editing this submission.

Line 104:  Change “sawn” to “sown”

Lines 107 to 112:  Provide a reference for this methodology.

Line 131:  There is no Wang et al. 2014 citation in the reference list.

Line 134:  See comment for line 100.

Line 154:  There is no Liu et al. 2016 citation in the reference list.

Line 170:  See comment for line 131.

Line 175:  There is no Zhang et al. 2020 in the reference list.  Do you mean “Zhao et al. 2020”?

Wording changes need to be made as per above section.  Could not complete review due to lack of line numbers.

Author Response

I did all of the grammar corrections according to your comments as you can see in the revised manuscript.

I added all the missing references in the revised manuscript.

Reviewer 2 Report

The authors analyzed 357 elite registered varieties (genotypes) of soft wheat grown in Europe over 3 years of research to identify candidate genes associated with grain protein content and how it can be implemented to improve its and end-use quality. This work is a continuation of similar genome-wide association studies by these authors. The authors were able to achieve their goal of identifying 35 genes (with high confidence). Although the authors write L.178. that "Protein content of grain was significantly dependent on years, genotypes, ......".  

The authors should specify in the Methodology that wheat genotypes are hexaploid. It is desirable, but not necessary, to specify how many soft wheat varieties belong to the spring form and how many to the winter form.

L. 46.  In the Latin species name, the letter "L" need not be italicized.

L. 108-112. To set forth the substance of the method of protein determination--without reference to the authors of the method.

L. 125. Where is the formula taken from, who is its author? It is necessary to give a reference to the primary source. 

The work is mainly theoretical in nature, based on the use of data from various databases (except for the determination of protein content in grain), software, genotyping.

Taking into account the title of the journal and its specificity, the introduction should include: 1. chemical composition of wheat grain; 2. forms of soft wheat - winter and spring; 3. expand the information on the economic importance of wheat.

Author Response

Thank you very much for taking the time to review this manuscript. Please find the detailed responses below and the corresponding revisions/corrections highlighted/in track changes in the revised manuscript.

I did all of the requested corrections according to your comments as you can see in the revised manuscript.

I added the references of the methodology and the formula in the revised manuscript.

I added more information in the introduction about the chemical composition of the wheat grains and its economic importance.

Reviewer 3 Report

This study sheds light on the vital role of wheat in providing calories and vegetable protein, especially in regions where animal protein is scarce. It focuses on grain protein content (GPC) and its impact on the quality of bread and baked goods. Through a Genome-Wide Association Study (GWAS) involving 367 European wheat genotypes over three years, the study delves into the genetic aspects of GPC.

Key findings highlight significant GPC variations influenced by genotypes, years, and their interactions, with some genotypes demonstrating notably high or low GPC levels. Notably, the study reveals specific candidate genes that play significant roles in GPC regulation, offering valuable insights for wheat breeding programs aimed at improving grain quality and nutritional value.

In summary, this study offers important insights into the genetic basis of GPC in wheat, with potential implications for enhancing wheat varieties to address global food security and nutritional challenges.

I am recommending this manuscript for publication. 

Author Response

Thank you very much for taking the time to review this manuscript and for recommending this manuscript for publication. 

Round 2

Reviewer 2 Report

The authors took into account all the reviewer comments and made appropriate changes to the manuscript. There are 2 small editorial changes:

The abstract should indicate the Latin name of the wheat species.

L. 48. Remove italics from L. Should be straight.

L. 85. Replace [22,23,24,25] with [22-25];

L. 338. Replace [17,22,23,24,25,26,27,28] (). at [17,22-28]. Remove parentheses.

The font throughout the text should be Times.